# Mycelium chemistry differs markedly between ectomycorrhizal and arbuscular mycorrhizal fungi

Weilin Huang [1,2 ✉], Peter M. van Bodegom [1], Stéphane Declerck[3], Jussi Heinonsalo[4,5], Marco Cosme[3], Toni Viskari[4], Jari Liski[4] & Nadejda A. Soudzilovskaia[1,2]

The chemical quality of soil carbon (C) inputs is a major factor controlling litter decomposition and soil C dynamics. Mycorrhizal fungi constitute one of the dominant pools of soil microbial C, while their litter quality (chemical proxies of litter decomposability) is understood poorly, leading to major uncertainties in estimating soil C dynamics. We examined litter decomposability of arbuscular mycorrhizal (AM) and ectomycorrhizal (EM) fungal species using samples obtained from in vitro cultivation. We showed that the chemical composition of AM and EM fungal mycelium differs significantly: EM fungi have higher concentrations of labile (water-soluble, ethanol-soluble) and recalcitrant (non-extractable) chemical components, while AM fungi have higher concentrations of acid-hydrolysable components. Our results imply that differences in decomposability traits among mycorrhizal fungal guilds represent a critically important driver of the soil C cycle, which could be as vital as is recognized for differences among aboveground plant litter.

[1] Environmental Biology, Institute of Environmental Sciences, Leiden University, Einsteinweg 2, 2333CC Leiden, the Netherlands. [2] Centre for Environmental Sciences, Hasselt University, Martelarenlaan 42, 3500 Hasselt, Belgium. [3] Mycology, Applied Microbiology, Earth and Life Institute, Université catholique de Louvain, 1348 Louvain-la-Neuve, Belgium. [4] Climate System Research, Finnish Meteorological Institute, 00101 Helsinki, Finland. [5] Department of Microbiology and Institute for Atmospheric and Earth System Research (INAR)/Forest Sciences, University of Helsinki, 00014 Helsinki, Finland. ✉email: w.huang@cml.leidenuniv.nl

The soil carbon (C) cycle is a critically important process for both ecosystem functioning and mitigation of climate change[1,2]. A major knowledge gap in this field is the lack of data on belowground influxes of C, and their fate in terms of contribution to stable C pools[3,4]. A particularly poorly understood aspect is the magnitude of C input into the soil pool of potentially decomposable C components as provided by belowground organisms, and the decomposability patterns of these organisms[5]. The chemical proxies of below-ground organisms' litter decomposability (referred to as "decomposability" hereafter), for example, the amounts of components extractable with various extraction techniques, of belowground organisms are among the key factors that influence the soil C turnover process[6–8]. The decomposability of substrates could influence the growth efficiency of new microbial biomass[9] and arguably mediate the ultimate fate of soil C, i.e., to be sequestered or respired[10]. However, due to the large uncertainty about the contribution of belowground organisms and their decomposability, until now, the largest known source of variability in the decomposability of C inputs into the soil has been associated with differences among plant species in terms of aboveground plant litter decomposability[11–13]. However, as soil organisms are excluded from these assessments we might be ~~an~~ underestimating ~~of~~ the true variability in the decomposability of C inputs.

Our knowledge about the factors that control the decomposability of C compounds entering into the soil pool through residues of microorganisms, especially so from widespread soil-borne fungi, is extremely limited. In soil ecosystems, mycorrhizal fungi living in symbiosis with plant roots are among the key soil microorganisms controlling the exchange of C and nutrients between soil and plants[14,15]. The living and dead biomass of these microorganisms constitute one of the most dominant pools of soil microbial C[16,17]. Depending on the soil ecosystem environment and mycorrhizal type, mycorrhizal hyphal biomass can constitute up to half of the standing mycelial biomass[18] and one-third of total microbial biomass[16]. Mycorrhizal fungi are important C sinks of net primary production (NPP)[19,20], and depending on the mycorrhizal guild, the annual mycelial accumulation can reach around 175–200 g C m$^{-2}$ [21,22]. This is particularly evident in some forest ecosystems, where the allocation of photosynthesized C into fungi can represent up to 30% of the NPP[21,23]. Yet, the magnitude of the potential contribution of mycorrhizal fungal pools to long-term soil C storage is unknown. Hence, a quantitative assessment of the chemical composition of microorganisms relevant for assessing the decomposability of microbial necromass is critically needed to narrow down the uncertainties in estimating belowground contributions to soil C pools[24,25].

Among the four principal types of mycorrhiza, the two globally dominant ones are arbuscular mycorrhiza (AM) and ectomycorrhiza (EM)[26]. These soil fungi associate with the roots of most terrestrial plants[26], and are predominant across the majority of the terrestrial vegetated areas[27]. Ecophysiological traits of these two main guilds of mycorrhizal fungi differ in many aspects (e.g., in the ability of enzymatic degradation of organic matter)[28,29]. Also, the microscopic structure of AM and EM fungal hyphae differs[5]. EM fungal hyphae have thicker walls, pigmentation, and septa between cells, and are generally believed to have a longer life span than AM fungal hyphae[17,30]. These differences in morphology could potentially determine the decomposability of EM and AM fungal litter, and have raised the hypothesis that dead EM hyphae are likely more recalcitrant to decomposition than AM hyphae[5]. However, our knowledge about the chemical differences among mycorrhizal fungal guilds, particularly on the chemical components that contribute differently to necromass decomposition, is remarkably limited.

Thus far, studies of the impacts of decomposition of fungal mycelium on soil organic matter (SOM) have focused primarily on EM and ericoid mycorrhizal fungi[31], and have examined mostly the abundance of individual chemical components in the fungal mycelium, such as concentrations of nitrogen, chitin and melanin. The latter is known to be negatively correlated to necromass decomposition of mycorrhizal fungal biomass[10,32]. While the outcomes of these analyses shed new light on eco-physiological traits of mycorrhizal fungi at the individual level, they (1) do not provide a comprehensive assessment of the potential fate of fungal biomass in the process of organic matter decomposition, and (2) neglect the most ancient and widespread mycorrhizal fungal guild, currently associated with the largest part of Earth's terrestrial vegetation – the AM fungi[26,27].

The objective of this study was to fill a main knowledge gap in the soil C cycle by examining inherent differences between EM and AM fungi in terms of their ultimate decomposability potential. Similar to plant litter residues that have a variety of components that differ in recalcitrance[33,34], variability of which among different types of plants, for instance among plant functional groups, have been recognized as the major factor controlling soil C dynamics for decades[11–13], soil fungi also consist of components of distinct decomposability[25]. Most fungi contain relatively recalcitrant components, such as melanin, that require costly oxidative enzymes for further decomposition[35], as well as relatively labile components (e.g., chitin) that are utilized as a source of C and N for the soil microbial community[36]. Upon release, these components enroll in principally different types of physical and chemical interactions with mineral surfaces and soil aggregates[3,37]. Yet the comprehensive understanding of principal differences among EM and AM fungi in terms of their decomposability is lacking.

An important reason underpinning this knowledge gap is the need for samples of in vitro pure biomass of mycorrhizal fungi to examine their chemical composition. For the AM fungi, this constitutes a particular challenge due to their obligate symbiotic lifestyle, which requires a suitable host root established on a poor medium to avoid any contamination by unwanted microbes. Using unique methods of cultivation of mycorrhizal fungi[38–40] established in the laboratory of mycology of the UCLouvain (Belgium), we cultivated multiple species of AM fungi under in vitro culture conditions and obtained amounts of fungal mycelia sufficient to examine their chemical compositions. To assess the differences in chemical traits between AM and EM fungal mycelium, we also cultivated EM fungi in vitro following standard laboratory techniques[41]. We subsequently assessed the chemical recalcitrance of AM and EM fungal mycelium. With this dataset, we tested two hypotheses crucial to understanding the contribution of major mycorrhizal fungal guilds to the soil C cycle:

(1) AM and EM fungal guilds differ principally in their chemical composition traits relevant for decomposability.
(2) Differences of decomposability between AM and EM fungal guilds are larger than the differences among litters from distinct plant functional types.

## Results and discussion

**Distinct chemical composition of AM and EM fungi.** We used 11 species of EM fungi and 4 species of AM fungi from in vitro cultures (CBS/MUCL number see Supplementary Table S1): EM fungal species are *Xerocomus rubellus*, *Paxillus involutus*, *Laccaria bicolor*, *Inocybe rimosa*, *Hebeloma hiemale*, *Lactarius deliciosus*, *Phaeogyroporus sudanicus*, *Peziza varia*, *Cortinarius cristallinus*, *Peziza quelepidotia*, and *Scleroderma verrucosum*; AM fungal species are *Rhizophagus clarus*, *Rhizophagus irregularis*, *Glomus aggregatum* and *Glomus hoi*. Samples of dried fungal biomass

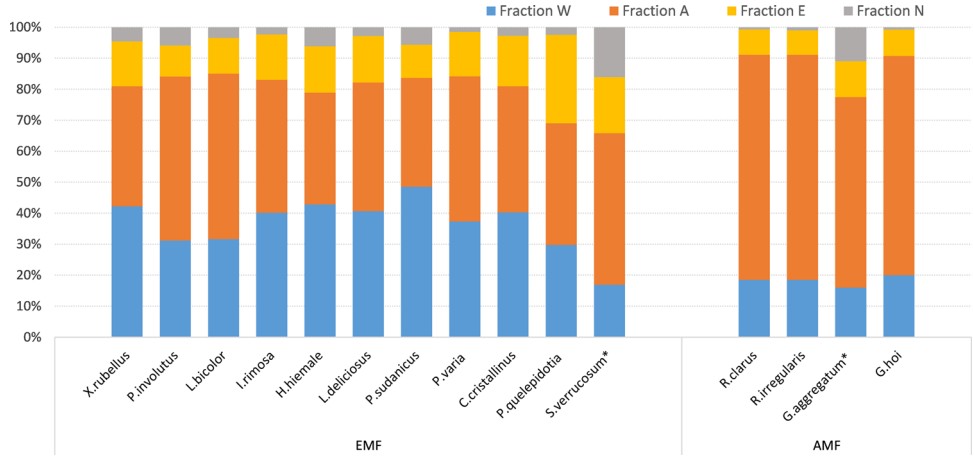

**Fig. 1 Chemical fractions of mycorrhizal biomass.** Relative abundance of water-soluble (W), acid-hydrolyzable (A), ethanol-soluble (E) and non-extractable (N) components in AM and EM fungi. Mycelia of *S.verrucosum** and *G.aggregatum** were assessed in a mixture with plant litter (details see Methods section).

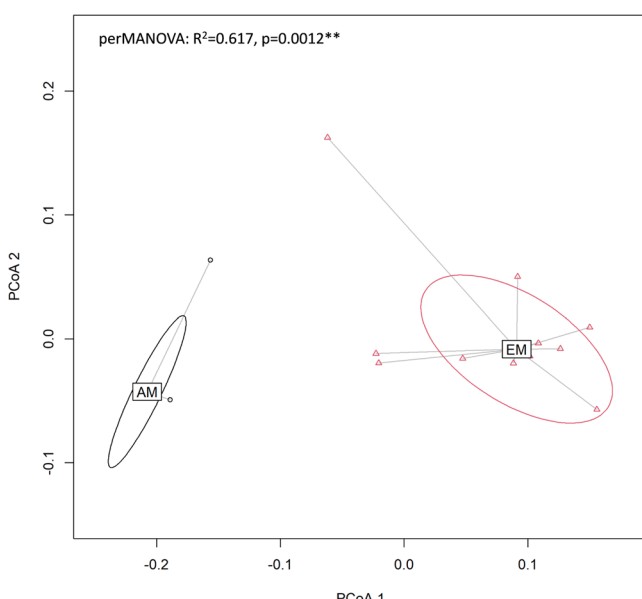

**Fig. 2 Principal coordinate analysis of mycelium chemistry.** Clustering and centroids of AM and EM mycorrhizal fungal species in a multidimensional space of WAEN components.

were examined for water-soluble, acid-hydrolyzable, ethanol-soluble, and non-extractable components (hereafter W, A, E, N components, respectively), allowing direct comparison of AM and EM fungi for the entire suite of recalcitrance traits (Fig. 1).

We found that the biomass of AM fungi exhibits a distinct set of decomposability-related traits compared to that of EM fungi (outcomes of a perMANOVA test on the WAEN components: $R^2 = 0.617$, $p = 0.002$, df $= 1$, for more details see Supplementary Table S3; dispersions of beta diversity, $p = 0.294$). In this analysis, data for individual fungal species were treated as replicates within AM ($n = 4$) and EM ($n = 11$) fungal guilds. The unambiguous difference between centroids of AM and EM fungi in the multidimensional space of WAEN components (Fig. 2) suggests that these two groups of fungi are likely to contribute to different pathways of soil C transformations as being direct sources of soil C compounds.

**Analyses of individual chemical compositions.** Subsequently, we examined which individual WAEN components differ

among AM and EM fungi. We found that concentrations of the most easily decomposable component (W), the ethanol-soluble fraction (E) and the most recalcitrant component (N) were significantly higher in EM fungi (Mann–Whitney tests with data for individual fungi within a fungal guild treated as replicates; $p < 0.001$, $p < 0.001$, and $p < 0.05$, respectively; Fig. 3a,c, and d). In contrast, AM fungi had a significantly higher A fraction (Mann–Whitney test conducted in the same manner as for W, A, and N components: $p < 0.001$; Fig. 3b). While both A and E components are of intermediate recalcitrance, acid-hydrolysable components have been shown to have higher decomposability than ethanol-soluble components[42]. The higher relative amount of E and N components in EM fungi compared to that of AM fungi constitutes novel empirical evidence in support of previous suggestions[24,43,44] that per fungal biomass units of EM fungi provide an important contribution to the soil pool of intact or partially oxidized mycorrhizal fungal biomass. At the same time, the high abundance of most easily decomposable W components supports empirical evidence of the high rate decomposability of EM fungal mycelium during the initial stages of decomposition[45,46].

Hereto we compared four plant functional types: evergreen trees, deciduous trees, evergreen shrubs, herbaceous plants. The differences between AM and EM mycorrhizal fungal species in chemical recalcitrance of litter, measured as the relative abundance of WAEN components, was nearly twice as large as the differences in chemical recalcitrance of litter between plant species of distinct functional types. For the particular case of the water-soluble components, it was even three times higher (Table 1). The effect sizes ($\eta^2$) of the difference between mycorrhizal fungal guilds for each individual WAEN component were higher. This was particularly evident for the major components of W and A (see Fig. 1) in the mycelium, which comprised the majority of the variation in chemical components. In contrast, a similar analysis conducted for plant species grouped into functional types (for details see Methods section) showed that the effect size of chemical differences of plant functional types was much smaller, and only acid-hydrolysable components contributed to the major variation in the group. Taken together, this suggests that the potential contribution to distinct pathways of C transformations differs markedly between mycorrhizal fungal guilds, and that differences in the decomposability pathways of mycorrhizal fungal material are even more striking than the differences observed in leaf litter among plant functional types, till now considered as one of the most important factors

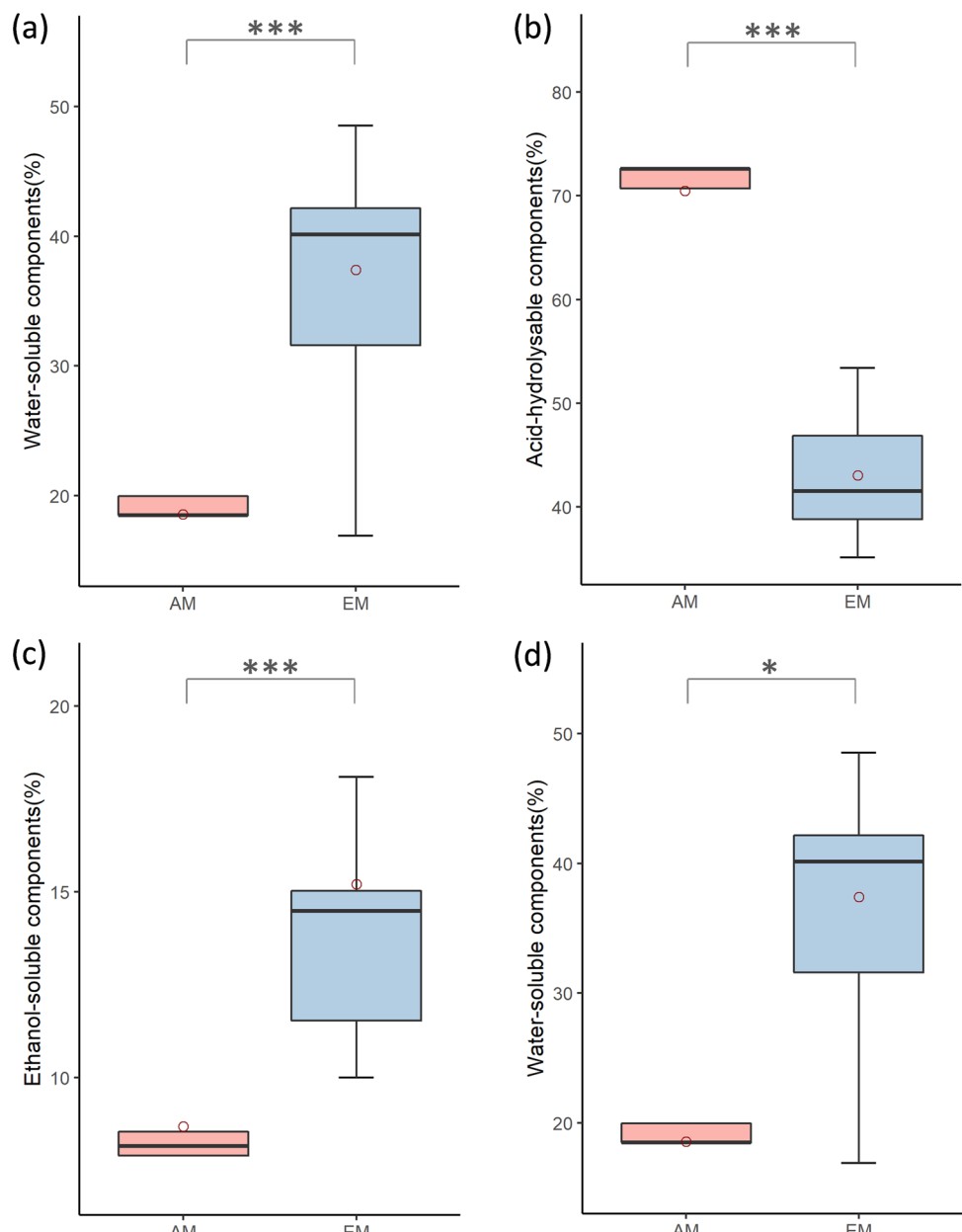

**Fig. 3 Boxplots of WAEN components in AM and EM fungi. a** Water-soluble components, $P < 0.001$, **b** Acid- hydrolysable components, $P < 0.001$, **c** Ethanol-soluble components, $P < 0.001$, **d** Non-extractable components, $P < 0.05$. All $P$ values refer to the Mann–Whitney test, AM ($n = 4$) and EM ($n = 11$). Upper and lower limits of the box- quartiles around the weighted-mean, horizontal lines within boxes- weighted-median values within each mycorrhiza group, and red circles - weighted-mean of each mycorrhiza group.

**Table 1 Comparing effect sizes of WAEN conposition differences in plant litter and mycorrhizal litter.**

|  | Water-soluble | Acid-extractable | Ethanol-soluble | Non-extractable |
|---|---|---|---|---|
| Mycorrhizal fungal guilds | 0.68 | 0.76 | 0.46 | 0.26 |
| Plant functional types | 0.21 | 0.41 | 0.28 | 0.20 |

Effect sizes (η2, one-way ANOVA) of chemical composition differences in leaf litter within plant functional types and mycelial biomass within mycorrhizal fungal guilds.

determining soil C circulation. Thus, chemical differences between mycorrhizal fungi types might be essential underestimated sources of (variation in) below-ground soil C dynamics.

Our test of chemical recalcitrance of mycorrhizal mycelium biomass of multiple EM and AM fungal species provides the first empirical evidence of the inherent difference between AM and EM fungi in terms of their chemical composition related to the decomposition pathway. Our study focussed on members of the globally most predominant family - Glomeraceae[47]. It remains to be determined whether less frequent AM fungal families with different life-history strategies potentially differ in their chemical composition.

Differences between AM and EM fungal guilds in decomposability support and mechanistically underpin previous speculations that EM fungi might contain a higher ratio of components recalcitrant to decomposition than AM fungi[5]. While microbiologists seek to specify the fungus-specific macromolecular compounds and basic chemical elements, these characters are difficult to link to soil C cycle mechanisms. Instead of analysing individual chemical components or complex chemical compounds of fungal biomass that are possibly a proxy for decomposability[5,10,32], we opted to characterize fungal biomass through general traits of litter decomposability known to drive soil C cycling[42,48,49]. Recently, it has been suggested that labile and recalcitrant C compounds originating from decomposing organic matter might follow distinct pathways of stabilization depending on the abundance of soil saprotrophic organisms[50,51]. This suggests that C components originating from mycorrhizal fungi of distinct guilds are likely involved in distinct pathways of C transformations in soil. Moreover, through the differential release of labile and more recalcitrant C components, the temporal dynamics of contributions of different mycorrhizal fungal guilds to distinct soil C transformation pathways will also differ among EM and AM fungi.

For decades, foliage litter and its variability among species or plant functional types has been considered as one of the main factors controlling soil C cycle process[11]. Our analysis shows that the magnitude of differences in decomposability traits between fungi of distinct mycorrhizal guilds is much higher than that of the leaf litter of plant species belonging to distinct functional groups. This suggests that the decomposability of mycorrhizal fungal biomass is a critically important factor for pathways of soil C transformation processes. Such pathways have been previously hypothesized[5,25] but in practice neglected or underestimated due to the high uncertainty associated with this phenomenon. This comparison of decomposability between mycorrhizal fungi and plant foliage litter is the first attempt at examining such characters of different substrates of decomposition litter. Future research with more fungal species may reveal information beyond the limited data used in this study. Given that plants allocate a significant part (up to 30%) of NPP to mycorrhizal fungal biomass[21,23], an amount comparable to the allocation into plant leaves in some ecosystems[52,53], the differential contributions of mycorrhizal fungal guilds to the processes of soil C turnover should be considered as a critical SOM formation factor. As mycorrhizal fungal necromass is among the most important sources of below ground soil C input, our results provide decomposability information of soil C inputs which is essential in narrowing down major uncertainties in estimating soil C fluxes dynamics.

## Methods

**Cultivation of AM and EM fungi**. We selected available AM fungal species strains from the Glomeraceae family, as this family is globally the most dominant family of AM fungi[47], while they can be grown in vitro producing reasonably large amounts of fungal biomass. We selected EM fungal species to cover relatively abundant strains of various families. In addition, we opted to use a higher number of EM fungi species compared to AM fungi, because (1) EM fungi consist of ca. 20,000–25,000 species[54,55] which entail high diversity of chemical traits, while AM fungi have been known to exhibit lower diversity with ca. 300 species identified within this fungal phylum[56–58], (2) mass-production of AM fungi to reach the amounts of biomass necessary for the recalcitrance assessments is complicated, necessitating hundreds of Petri plates. We have opted to use laboratory cultivation protocols adapted to each fungal species to assure that each species develops in a most healthy way during mass cultivation. This ensures that the chemical composition of fungal material examined is representative for each mycorrhizal type in general, while it may differ from specific local soil conditions. Through cultivation, all manipulations were conducted under sterile conditions to prevent contamination of fungal material, by using a laminar flow hood, and with sterile or sterilized laboratory material.

**EM fungi cultivation and sample preparation**. Original cultures of EM fungal species were obtained from Westerdijk Fungal Biodiversity Institute (the Netherlands), which also provided standard laboratory instructions for EM fungi cultivation (except strain of *Scleroderma verrucosum*, which was obtained from the collection of GINCO). We assumed that the chemistry of fungi cultivated following standard cultivation protocols is only affected by the species morphology. Each species was inoculated in 30–80 Petri plates (90 mm, diameter), containing species-specific medium (Supplementary Table S1) solidified with bacteriological agar, then sealed with film and incubated in climate rooms (temperature 21–27 °C according to the preference of each strain, in the dark) for 4–5 weeks (Supplementary Fig. S1). Harvested fresh mycelium of EM fungi was washed with distilled water for 10 s, collected by filtration, and stored at −20 °C. The frozen fungi biomass samples were dried using a freeze dryer or oven under 55 °C for at least 12 h (weighed after another 4 h until the weight is stable, drying methods see Supplementary Table S1), then stored at −20 °C before chemical recalcitrance assessments.

**AM fungi cultivation and sample preparation**. All AM fungal strains were obtained from the Glomeromycota in vitro Collection (GINCO, Belgium). The cultivation protocol of AM fungi followed the methods well-established in the laboratory of mycology of UCLouvain (Belgium). As AM fungi are relatively slow-growing, and there was no prior knowledge on biomass output among in vitro cultivation approaches. We recruited a combination of different cultivation characteristics to maximize biomass productivity, and each strain was cultivated using four different systems (Fig. 4): autotrophic whole plants system either with a Petri (S1) or a mesh (S2) root compartment (RC), transformed root organ culture (ROC) system in bi- (S3) or mono-compartmented (S4) Petri plates. In the end, we established over 600 AM systems, all biomass produced by each strain in the four different in vitro systems described below was needed to fulfill the standard amount required for the chemical analysis.

System S1 consisted of a lid of a small 50-mm-diameter Petri dish placed inside a large 145-mm-diameter Petri dish, to create an RC inside a mycelial compartment (MC) (Fig. 4c). System S2 was similar to S1 with the difference that a 55-mm-diameter cap made with 40 μm nylon mesh and filled with cotton was used as RC instead (Supplementary Fig. S2). The RC contained roots of mycorrhizal plants to sustain fungal growth into the MC. Both compartments were filled with modified Strullu–Romand (MSR[59]) medium without sucrose and vitamins. The large plates were covered with black plastic foil to minimize light exposure. In each large plate, the plant shoot grew outside through a 2-mm-diameter lateral opening sealed with sterile silicon grease as described[60]. The systems were kept in a growth chamber with a 16 h photoperiod, 130 μmol m$^{-2}$ s$^{-1}$ light intensity, 27 °C temperature, and 80% relative humidity. Each RC was refilled with medium every 2–3 weeks.

System S3 (Fig. 4b) consisted of 94-mm-diameter bi-compartmented Petri plates with RC and MC. The RC contained mycorrhizal Ri T-DNA transformed roots clone DC2 of *Daucus carota* growing in MSR medium to sustain the fungal growth into the MC. The MC was filled with MSR medium without sucrose and vitamins. These bi-compartmented plates were incubated inverted in the dark at 27 °C for 6 months; The system S4 (Fig. 4a) consisted of a 145-mm-diameter mono-compartmented Petri plate with mycorrhizal Ri T-DNA transformed roots

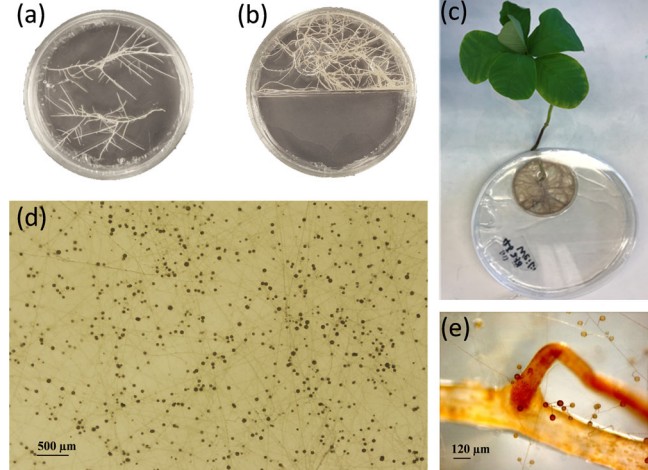

**Fig. 4 In vitro cultivation of arbuscular mycorrhizal fungi.** Cultivation system in (**a**) mono-compartmented or (**b**) bi-compartmented Petri dish on excised Ri T-DNA transformed root organs of carrot. **c** Cultivation with the whole plant of Crotalaria Spectabilis in a bi-compartment Petri dish system; **d** Mycelium and spore production of the AM fungus (*Rhizophagus irregularis*, MUCL 41833) grown in a bi-compartmented Petri dish and (**e**) a mono-compartment Petri dish.

of *D. carota* clone DC2 growing in MSR medium[59]. The plates were incubated inverted in the dark at 27 °C for 4–5 months.

For systems S1, S2, and S3, roots were trimmed before invading the MC to keep the MC root-free. Once the MC was full with mycelium, the medium was harvested to extract the mycelium as described below, and the MC was re-filled with medium to allow fungal re-growth. The harvesting procedure was repeated for each plate every 4–6 weeks until another 6–10 months according to the productivity of each plate; For system S4, only the sections of the medium without any roots were harvested once to exclude roots and root exudates after incubation.

The absence of roots in the harvested medium was carefully evaluated and confirmed using a stereomicroscope. For all different systems and strains, the harvested medium which only contained mycelium was immediately liquefied inside a beaker in a water bath at 70 °C for 2 h—this procedure also killed the mycelium. The mycelium was then collected using a 38-μm filter, washed with demineralized water for 10 s to remove any remnants of medium and root exudates (only possibly exist in the harvested medium from S4), and stored at −20 °C until further use. Prior to chemical analyses, all mycelia were dried using the same procedure as for the EM fungi described above (Supplementary Table S1).

**Decomposability analysis**. Chemical composition of mycelia was examined in the laboratory of Natural Resources Institute (Finland). We examined the fungal samples for different extractable components based on their solubility in distinct solvents which has been widely used in investigating compositions of plant litter and soil organic matter[61]: The W-fraction is largely composed of carbohydrates and nitrogen-containing compounds, and is frequently used as a measure for potentially bioavailable SOM and thus for the readily decomposable C pool[62]; The A-fraction: acid hydrolysis can extract carbohydrate and protein materials by disruption of hydrolytic bonding, leaving the more biologically recalcitrant alkyl and aryl materials largely intact[63,64]; The E-fraction: dichloromethane was used for extracting nonpolar extractives[65] (e.g., fatty acids, long-chain alcohols, wax esters, oils, resins, etc.); The N-fraction is the residue remaining after hydrolysis in sulfuric acid (also known as Klason lignin). The amounts of extractable substances were determined gravimetrically by incubating samples with a solvent and weighing the samples after filtration and drying. Mass loss during each extraction was considered to be equal to the amount of a compound being extracted. For details of the protocols, see Ryan et al. (1990)[66] and Wieder and Starr (1998)[67].

The WAEN components extracted through different chemical methods represent the suite of decomposability traits[61] related to the extraction capacity of different types of enzymes potentially excreted by saprotrophic organisms[68]. These components with different decomposability are key to determining the dynamics of litter decomposition and soil C cycling in soil C modeling[42,48,69]. The decomposability order W-A-E-N was determined based on the Yasso soil C model[42,48], which is coherent with the findings of real plant litter decomposition experiments (see Supplementary Fig. S3).

The raw measurement results corresponding to Fig. 1 are provided in Supplementary Table S2. Samples that did not reach 0.5 g were measured with a mixture of plant litter (with a known content of WAEN components) to reach the necessary quantity for analysis, and the fungal chemical composition ($G_i$) was calculated based on the proportion of fungal biomass in the samples according to the following equations:

$$W_T = W_f + W_p \tag{1}$$

$$G_i = (W_T - G_i{}' \cdot (W_p \cdot W_T{}^{-1})) \cdot (W_f \cdot W_T{}^{-1}), i \in \{W, A, E, N\} \tag{2}$$

where $W_T$ is the total weight of the mixture of dried plant litter and fungi; $W_p$ is the weight of dried plant litter; $W_f$ is the weight of dried fungal biomass; $G_i{}'$ is the measured value of each chemical composition (WAEN) in the mixture.

We calculated the WAEN fractions for fungi and estimated the corresponding accuracy with different portions of fungi and standard litter mixture (Table 2). Based on the relatively high proportion of N-fraction in standard plant litter, which increases the uncertainty in N-fraction estimates in mixtures, data for fungal species assessed in a mixture with plant material was assigned a lower weight (0.5 instead of 1 as default weight value for other data) in the data analysis (next section).

**Statistics and reproducibility**. We assessed the significance of the overall differences in recalcitrance between AM and EM fungi, with the permutational analysis of variance – perMANOVA[70], performed with 999 permutations in the Vegan package (Bray–Curtis function) in R. Data distribution and homogeneity of variance of original WAEN values are provided in Supplementary Figs. S4, S5 (include the information about log-transformed values). Dispersions of beta diversity (the distance from an individual measure to the group's centroid) were calculated for each beta diversity metric within AM and EM fungal groups for estimating within-group variation across individuals. Significant differences in beta diversity variation[71] were tested using permutational statistical tests for the homogeneity of group dispersions with 999 permutations in Vegan. We used Principal Coordinates Analysis (PCoA) for visualization of the data present in the beta diversity distance matrix (Fig. 2).

Subsequently, we tested the hypothesis that AM fungi exhibit higher amounts of easily soluble, and acid-hydrolysable compounds, while EM fungi have higher

**Table 2 Accuracy assessment for experiments measured with mixture.**

| Mycelium proportion in the mixture with leaf litter | Mixture AWEN value from experiment | | | | Mycorrhizal AWEN fraction from the calculation | | | |
|---|---|---|---|---|---|---|---|---|
| | W' (mix) | A' (mix) | E' (mix) | N' (mix) | W | A | E | N |
| (0 g fungal mycelium + 0.5 g leaf) 0% | 9.29 | 10.60 | 45.81 | 34.30 | \ | \ | \ | \ |
| (0.10 g fungal mycelium + 0.40 g leaf) 20% | 15.32 | 11.76 | 42.99 | 29.92 | 39.46 | 16.40 | 31.73 | 12.42 |
| (0.20 g fungal mycelium + 0.30 g leaf) 40% | 23.65 | 13.38 | 38.41 | 24.56 | 45.18 | 17.55 | 27.31 | 9.96 |
| (0.30 g fungal mycelium + 0.20 g leaf) 60% | 33.25 | 14.90 | 33.51 | 18.34 | 49.23 | 17.77 | 25.31 | 7.69 |
| (0.40 g fungal mycelium + 0.10 g leaf) 80% | 42.70 | 16.18 | 29.66 | 11.45 | 51.06 | 17.58 | 25.62 | 5.74 |
| (0.50 g fungal mycelium + 0 g leaf) 100% | 51.86 | 17.95 | 26.05 | 4.14 | 51.86 | 17.95 | 26.05 | 4.14 |

Accuracy assessment of the fungal chemical composition calculated from WAEN test results of a mixture with plant litter. EM Fungi of *Hebeloma hiemale* biomass and a typical northern conifer tree branch were used for this estimation.

amounts of compounds that are neither soluble nor hydrolysable, by a non-parametric Mann–Whitney U test (with package *sjstats* in R) to determine if there were statistically significant differences in each chemical component between the two mycorrhizal groups. To account for the fact that WAEN of two fungal strains were assessed in a mixture with plant litter, all statistical analyses of fungal WAEN were conducted as weighted analyses according to the accuracy assessment for the results of sample from a mixture.

We examined the magnitude of the difference between the recalcitrance of mycorrhizal fungal types vs. the recalcitrance of plant material (Hypothesis 2) comparing the effect size- Eta square ($\eta^2$)[72] of ANOVAs on WAEN values of AM vs EM fungi to the effect sizes of ANOVAs on WAEN of plant functional types (other effect size indices, dispersions, and variation across groups of plant litter are provided in Supplementary Note S1, Table S5, Fig. S6). To meet the normality assumptions, WAEN values were log-transformed.

The data on plant functional types used for this analysis were obtained as follows: We gathered plant leaf WAEN chemical composition data for 57 species from CIDET[33] and LIDET[34] datasets (details see Supplementary Note S1, Supplementary Table S4). Those data were grouped into the evergreen tree, deciduous tree, evergreen shrub, and herb, based on plant growth form information from the TRY database[73]. Species with multiple form definitions were defined according to the highest occurrence frequency[74].

**Reporting summary**. Further information on research design is available in the Nature Research Reporting Summary linked to this article.

## Data availability
All data generated or analyzed during this study are included in this article (and its supplementary information file).

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

## Acknowledgements

This research was supported by the Vidi grant 016.161.318 (issued to N.A.S. by The Netherlands Organization for Scientific research) and China Scholarship Council (CSC, grant No. 201706040071 issued to W.H.). M.C. was supported by the European Commission's grant H2020-MSCA-IF-2018 'SYMBIO-INC' (GA 838525). T.V., J.L., and J.H. were supported by the Strategic Research Council at the Academy of Finland (decision 327214, 327342) and the Nessling foundation TWINWIN project. We appreciate the Natural Resources Institute Finland and Prof. Hannu Fritze for supporting chemical analysis. We would like to thank colleagues of the Soil-process group and Chen Li (CML, Leiden University) for discussions. We also thank the anonymous reviewers for their constructive comments and suggestions.

## Author contributions

W.H., P.M.B., and N.A.S. conceived the original idea and planned the project. S.D. and J.L. were involved in planning the project. W.H. carried out the cultivation with assistance from S.D. and M.C. on AM fungi. J.H. provided the sample measurements and processed the experimental data. W.H. performed the numerical calculations and analysed the data. P.M.B, N.A.S, J.H., and T.V. aided in interpreting the results. W.H. and N.A.S. wrote the manuscript in consultation with P.V.B, S.D., M.C., J.H., T.V., and J.L. All authors discussed the results and commented on the manuscript.

## Competing interests

The authors declare no competing interests.
