## [Peer Review File · Communications Biology]

Reviewers' comments:

Reviewer #1 (Remarks to the Author):

This paper examined the mycelium chemical quality of ectomycorrhizal fungi and arbuscular mycorrhizal fungi with the purpose of determining differences in decomposability among guilds.

As any other culture-based study, the effort authors put in culturing was very impressive. They tested multiple different ECM and AM species and had good 'within' replication. Their methods are well-described although they seem to be somewhat complicated to replicate since the level of detail seems quite specific for each system.

Moreover, the results authors are reporting are important for quantifying the fungal necromass contribution to the carbon cycle, which at the moment is not included in carbon dynamics modelling and in fact, is unknown.

These types of studies show that the fungal community necromass contribution to the carbon may be guild-dependent, which is especially relevant considering there are specific-ECM and AM dominated ecosystems. The results these authors are reporting, may suggest that the carbon cycle in these different ecosystems may vary substantially depending on the type of fungal community dominance and necromass input.

Overall, I found this paper very interesting and easy to read. Results are also straight forward and clear. I have a few comments:

2-4 – confusing sentence. What do authors mean by “their litter quality”?

13 – The carbon cycle is one of the better understood biogeochemical cycles, compared to for example the phosphorus or manganese cycle. I do agree it is critically important for ecosystem functioning but definitely not one of the least understood.

111-113 – Please specify replication, so for ECM $n = 11$, and AM $n = 4$.

310 – Can authors elaborate on why they chose Mann-Whitney U test instead of a parametric test? They had already logged transformed the data -I assume to meet normality- so I'm confused as to why they chose a non-parametric test here.

315 – I consider this information to be important enough to be included in the main methods and not as supplementary information.

Reviewer #2 (Remarks to the Author):

In this study, chemical profiles of arbuscular (AM) and ectomycorrhizal (EM) fungi were compared and the difference between AM and EM fungal chemical compositions was compared to plant functional groups. Overall, I found the topic to be very interesting, and I am very impressed with the techniques used to generate the mycorrhizal fungal biomass which made this study uniquely possible. I also think the authors are testing something very novel and important, as the chemical differences across different types of fungi remain largely unknown, especially for AM fungi. The authors also suggest that these differences in chemistry are larger between AM and EM fungi than across plant functional groups. This is a major finding. Despite enjoying this paper, I have one general concern and a number of more specific concerns which I hope help to improve the paper.

Foremost, the use of rather classic extraction techniques and grouping different types of extracts into decomposability categories raises some concerns. While this is very common - or used to be - it is widely recognized that this is an operational way of understanding chemical groupings and is not based on ecological principles (e.g. Lehmann and Kleber, 2015, Nature; Dungait et al 2012, Global Change Biology). Even though these groups are used in carbon cycle models, making them

useful categories, there are concerning artefacts of these techniques. The biggest strength of using this approach is its direct comparability to data collected for plants. Using this connection to plant data plus the use of some of these values in some C cycling models would strengthen the rationale of using this method.

If the authors also decomposed the mycelial material, they would be able to link decomposition rates to these classes and make a more empirical case. Further, if the authors used more modern spectroscopic techniques, then they would be able to provide finer resolution details and draw clearer inferences to decomposability and adsorption onto minerals. While it is not my intention for them to do more work, I think they need to clearly state what each extraction group might mean and use examples from the literature as support.

Line-by-line comments

4: It might be good to define 'chemical recalcitrance' here or to use another word since this term is not universally accepted in the literature.

19: I would also define here what is meant by "quality" - as this could mean chemical recalcitrance or something else...

21: This is absolutely possible, but it could also be interpreted as lending support to the idea that chemical recalcitrance causes C accrual, which you obviously want to be careful about suggesting. It might be useful to cite some papers discussing how chemical recalcitrance of substrates influences microbial growth efficiency and in turn C storage (e.g. Frey et al. 2013, Nature Climate Change).

25-26: I very much agree.

53-54: Could you specify if this is something you are hypothesizing here or if this is a more general statement. If it is a more general statement, adding some citations would strengthen the argument.

Line 63: I'm not super clear what 'integrative' means...could you be more specific? This also applies to line 76.

Line 71: It might be better to use a different example from melanin here since not all fungi produce this.

Line 93-34: This hypothesis was somewhat surprising since you did not provide background information in the introduction on this comparison. I would add this background information in order to strengthen this point. This comparison is so interesting, and I really want your future readers to understand the potential significance of it.

101-102: I very much respect how hard it is to isolate and culture AMF - but only four species are used here and they are all Glomeraceae. Thus, your results about AMF are limited...it would be good to acknowledge this in the discussion. Other families which produce more extensive biomass in the soil (e.g. Gigasporaceae) or which might be more stress tolerant (e.g. Acaulosporaceae) could have quite different mycelial chemistry.

111: You also note an analysis of variance in the methods section - are those results not included?

133: Why do you think you discovered higher A components in AMF biomass? Could it have anything to do with the potentially higher spore volume in the AM vs. EM mycelium? It would also be interesting to acknowledge the species which stand out and why that might be.

142: I find this framing of mycorrhizal versus plants very interesting, and I like it, so I want others to appreciate it. To make it more generally accepted, I'd like to see you discuss any potential limitations of this comparison. Notably, how does variation in the sample size of plants versus fungi influence these results? If you do not think this has an effect, can you state why you think the results are robust to unequal sample sizes and potential unequal variance? Do you see the same results if you randomly sample from the plants to equal the sample size of fungi? Do you see something similar if you compare the groups using non-parametric techniques (e.g. comparing coefficients of variation across all groups) or multivariate analysis of dispersion? Not all of these need to be tested, I'm just proposing some ideas that could complement the findings beyond the parametric effect size measurements.

170-172: Your results do not entirely support this conclusion. The A compounds are most abundant, and there were higher concentrations in AMF versus EMF tissues. I would therefore be more cautious with this statement as it only applies in relation to the E and N compounds.

220: It would help to add a sentence here saying why or why not these fungi growing on different media would or would not affect the results. For example, differences in the media C:N ratio seems like it would differentially influence mycorrhizal biomass chemistry.

237: I would add a sentence here describing why these different methods were used. Since you say they are "well-developed" in the lab of UCLouvain, it seems unclear why all four approaches needed to be used.

291-292: I think additional details on these methods and interpretations need to be added to the methods and discussed very clearly in the discussion. As noted above, I particularly argue this because the concept of recalcitrance is centered in the manuscript. I am concerned that some of these methods rely on concepts which have very little biological basis (e.g., as discussed by Lehmann and Kleber, 2015, Nature). For example, these are operationally defined chemical fractions, some of which are well-known to contain artefacts from sample preparations (e.g., acid extractions and assuming the non-hydrolysable mass is very refractory is no longer widely accepted). Some of these artefacts have given rise to absurdly long residence times of certain types of compounds in plants (e.g. lignin) which actually decompose within 1 year (e.g., Rasse et al 2006, J. of Soil Science).

I encourage you to outline each extraction, and how it can be interpreted based on previous evidence. I would not rely on soil C models for this either since it is these very ways of grouping that has given rise to flawed estimates....many propose we need to move beyond this way of binning C inputs (e.g. Todd Brown et al. 2013, Biogeosciences; Sulman et al 2018, Biogeochemistry).

In the discussion, it seems important to acknowledge the limitations of using this classic extraction approach and why more modern spectroscopic methods were not used beyond the one stated sentence.

293: From Note S1 and Table S3, adding plant litter seems to have a huge effect that cannot be corrected for in all of the extractions, especially for the non-extractable compounds. Based on Figure 1, the two samples which were mixed with plant litter have the largest N columns...this seems like it is an artifact of the plant mixture. I'm concerned that these two cultures may not be directly comparable to the other species for this reason.

299: Which distance metric was used for these analyses? If Euclidean distance, were they independent?

301: Because the EM and AM sample sizes differ, was this used to test whether variation in PERMANOVA results was due to dispersion or differences in centroids (sensu Anderson and Walsh 2001, Ecological Monographs)?

310: Were these tests also in R?

312-315: If you exclude these two species do you detect the same results?

318: Is ANOVA appropriate given the fact that you used non-parametric tests for comparing between EM and AM samples? What makes parametric tests appropriate here but not earlier? I'm also a bit concerned by the larger sample size for plants versus fungi here. This unequal sampling seems like it would potentially affect within group sample variance? Would you consider trying to account for this by adding variance structure statements in your models (e.g. using the varIdent function R). Do you also have replicate values for fungi but not for plants? If you have replicate values for both fungi and plants you could also use mixed effects models and use species as a random effect.

Reviewer #3 (Remarks to the Author):

Huang and colleagues presented a study about the differences of mycelium decomposability between AM and EM fungi from samples obtained from in vitro cultivation. They found that mycelium decomposability of AM and EM fungi varies among different chemical decomposable substances, having EM fungi a higher concentration of labile and recalcitrant components than AM fungi. I found this manuscript interesting and fulfill a knowledge gap in the soil carbon literature. The experiments are simple and straight forward. However, there are some issues that needs to be addressed/clarified.

The first main comment I have is that the manuscript makes a strong emphasis on soils. Yes, I agree that EM and AM fungi are an overlooked component of the terrestrial carbon cycle, however, the mycelium used was grown and harvested from specific media in petri dishes (where is the

soil?). This issue has two components that the authors did not discuss: 1) there might be chemical differences between EM and AM fungi grown in media vs grown in soil? 2) those chemical differences might be translated to different decomposability of both EM and AM fungi. In this matter, there might be different solutions, but I suggest to the authors to acknowledge that the results are only "potential" decomposability rates, as well as that the chemical composition of the grown mycelium might be different than grown in actual soil. This might be a substantial change in the text to tone down the global biogeochemical implications.

The second main comment I have is about the comparison between the chemical decomposability traits between EM and AM fungi versus plant litter. I found this comparison out of place in the manuscript and should be removed. There are different reasons for this: 1) the title of the manuscript is about EM and AM fungi, and the decomposition of plant litter is just a final line in the abstract, with no substantial results; 2) in the introduction section it is only mentioned in the hypotheses (!), so there is not background on why this comparison is made or why this is important, and it is discussed in the results/discuss section; 3) data for the litter decomposition come from two different known databases, with different sample sizes for the plant functional types, making a no-fair comparison.

My third main comment is related to how the authors reported the sample sizes and how the statistics were done. 1) It is not clear which is the real sample size of EM and AM fungi for stats, seems that 600 for AM but where is the value of EM?; the authors used the fungal guilds as replicates ($n = 4$ for AM and $n = 11$ for EM), are the authors using an average value from averages values? If so, why not using the raw data pooled together as AM or EM? Using an average of averages smooths a lot of variability, and if the sample size is different the Mann-Whitney test might be still used. 2) differences presented in figure 3 (boxplots) come from the $\log(x+1)$ transformed of, again, $n = 4$ for AM and $n = 11$ for EM, arguing that the log transformation is due to lack of normality (are we expecting normality with an $n=4$?); I suggest to the authors to report the results from a normality test; however, I strongly suggest using the raw data and not using the fungal guilds as replicates. This should be also reflected in submitting all the raw data to a repository and not only reporting the "limited" data in the supplementary material. Another potential solution could be reporting in supp mat boxplots of the different fungal guilds so the reader can see that there is a difference between EM and AM fungi by guilds.

Feedback to reviewers' comments

Dear reviewers,

Thank you very much for your appreciation of our manuscript and the constructive comments and suggestions. In the table below, we have addressed them one by one. The line numbers in the right table column refer to the lines in the revised manuscript (marked-up version). To ease the reviewing we have highlighted all the changes in the manuscript file with blue color.

With best wishes,

Weilin Huang and co-authors

NO.	Remarks to author	Replies and modifications
Reviewer #1 _R1.0.	This paper examined the mycelium chemical quality of ectomycorrhizal fungi and arbuscular mycorrhizal fungi with the purpose of determining differences in decomposability among guilds. As any other culture-based study, the effort authors put in culturing was very impressive. They tested multiple different ECM and AM species and had good ‘within’ replication. Their methods are well-described although they seem to be somewhat complicated to replicate since the level of detail seems quite specific for each system. Moreover, the results authors are reporting are important for quantifying the fungal necromass contribution to the carbon cycle, which at the moment is not included in carbon dynamics modelling and in fact, is unknown. These types of studies show that the fungal community necromass contribution to the carbon may be guild-dependent, which is especially relevant considering there are specific-ECM and AM dominated ecosystems. The results these authors are reporting, may suggest that the carbon cycle in these different ecosystems may vary substantially depending on the type of fungal community dominance and necromass input. Overall, I found this paper very interesting and easy to read. Results are also straight forward and clear. I have a few comments:	We thank the reviewer for the positive comments, support and quality of evaluation.
Reviewer #1 _R1.1.	2-4 – confusing sentence. What do authors mean by “their litter quality”?	We agree with the point that we need to clarify the definition of “litter quality”, “chemical quality” and “chemical decomposability”. In the sentence, “their litter quality” refers to the chemical proxies of litter decomposability (for example the water-soluble,

		acid-hydrolyzable, ethanol-soluble and non-extractable components in the litter). We added descriptions to explain the phrase (Line 3). Moreover, we ensured consistent choice of wording on this topic throughout the paper using 'decomposability'.
Reviewer #1 _R1.2.	13 – The carbon cycle is one of the better understood biogeochemical cycles, compared to for example the phosphorus or manganese cycle. I do agree it is critically important for ecosystem functioning but definitely not one of the least understood.	We agree that there are other biogeochemical cycles even less understood. Still, there are a lot of uncertainties and unknowns about the soil carbon cycle when we look at the global C circulation. Therefore, we have softened the sentence as follows: 'The soil carbon (C) cycle is a critically important process for both ecosystem functioning and mitigation of climate change' (Lines 14-16).
Reviewer #1 _R1.3.	111-113 – Please specify replication, so for ECM n = 11, and AM n = 4.	Thank you for pointing this out. We specified the number of replicates in the sentence "AM (n=4) and EM (n=11)" (Line 122).
Reviewer #1 _R1.4.	310 – Can authors elaborate on why they chose Mann-Whitney U test instead or a parametric test? They had already logged transformed the data -I assume to meet normality- so I'm confused as to why they chose a non-parametric test here.	Thank you for raising this point. Below is a description of how and why we decided to use the specific method for each analysis: First of all, we primarily aimed to investigate the general differences in chemical composition patterns between AM and EM fungi by conducting a multivariate analysis. We chose the non-parametric perMANOVA because it is: (1) less sensitive (as compared to e.g. ANOSIM and the Mantel) to differences in dispersions among groups; (2) it is not sensitive to differences in correlation structure (shape) among groups (Anderson, 2001, 2017). Subsequently, we also wanted to compare between individual components. Here we have also chosen a non-parametric method, because (1) our sample size of EM and AM was small and data was still not normally distributed after a logarithmic transformation; (2) this ensured consistency with the non-parametric perMANOVA. We tried to use the log-transformation to mitigate heterogeneity of variance before conducting analysis, but that did not solve the heterogeneity issues (we added this information into supplementary material, see Fig.S4). Thus we have redone these two non-parametric analyses using the original data. It did not influence our main

		results and conclusions. Please see updates in the methodology (Lines 346-348), Fig.2 and Fig.3.
Reviewer #1 _R1.5.	315 – I consider this information to be important enough to be included in the main methods and not as supplementary information.	We agree with this suggestion and moved this section to the methods. See update in the main methods in Lines 326-342. Thank you for your advice.
Reviewer #2 _R2.0.	In this study, chemical profiles of arbuscular (AM) and ectomycorrhizal (EM) fungi were compared and the difference between AM and EM fungal chemical compositions was compared to plant functional groups. Overall, I found the topic to be very interesting, and I am very impressed with the techniques used to generate the mycorrhizal fungal biomass which made this study uniquely possible. I also think the authors are testing something very novel and important, as the chemical differences across different types of fungi remain largely unknown, especially for AM fungi. The authors also suggest that these differences in chemistry are larger between AM and EM fungi than across plant functional groups. This is a major finding. Despite enjoying this paper, I have one general concern and a number of more specific concerns which I hope help to improve the paper.	We thank the reviewer for the recognition of the potential value of our work as well as for the detailed suggestions to improve the manuscript. We have addressed the comments as indicated below.
Reviewer #2 _R2.1.	Foremost, the use of rather classic extraction techniques and grouping different types of extracts into decomposability categories raises some concerns. While this is very common - or used to be - it is widely recognized that this is an operational way of understanding chemical groupings and is not based on ecological principles (e.g. Lehmann and Kleber, 2015, Nature; Dungait et al 2012, Global Change Biology). Even though these groups are used in carbon cycle models, making them useful categories, there are concerning artefacts of these techniques. The biggest strength of using this approach is its direct comparability to data collected for plants. Using this connection to plant data plus the use of some of these values in some C cycling models would strengthen the rationale of using this method. If the authors also decomposed the mycelial material, they would be able to link decomposition rates to these classes and make a more empirical case. Further, if the	We agree with the remarks. Also, please see our reply to R2.17. There are - as with every decomposability classification - concerns about its use and no doubt a decomposition experiment would have enhanced the research results, and we are planning and preparing to conduct an experiment on the decomposition rate of mycelial material in the future. The main reason we did not perform this in the current paper is that the amount of fungal biomass required for a decomposition experiment is extremely demanding to reach for the AM fungi (considering the time it took to get sufficient biomass for the current experiments). The WAEN extraction method creates measurable fractions based on their solubility in different solvents in different temperatures. Since most of the soil organic matter is decomposed through oxidative and hydrolytic enzymes/processes (Xu et al., 2020), the chemical extractions (especially acid

authors used more modern spectroscopic techniques, then they would be able to provide finer resolution details and draw clearer inferences to decomposability and adsorption onto minerals. While it is not my intention for them to do more work, I think they need to clearly state what each extraction group might mean and use examples from the literature as support.

and water extractions process) constitute a rough approximation of these processes. Thus, the components extractable through different chemical methods represents a suite of decomposability traits, representing components extractable through different types of enzymes potentially excreted by saprotrophic organisms. In this context, we use chemical fractions as a way to operationalize the decomposability of mycorrhizal fungi for which currently no generic alternative methods exist.

Although we don't know what each extraction represents for fungi exactly, this set of extractions has been widely used in investigating the compositions of plant litter and soil organic matter (von Lützow et al., 2007). We appreciate the criticisms on the interpretation of such measures and, as suggested by the reviewer, we deal with these by providing in the main the manuscript text explanations on the biochemical meanings of each fraction as follows: **The W-fraction** is largely composed of carbohydrates and N-containing compounds, and is frequently used as a measure for potentially bioavailable SOM and thus for the readily decomposable C pool (Davidson et al., 1987); **The A-fraction**: acid hydrolysis can extract carbohydrate and protein materials by disruption of hydrolytic bonding, leaving the more biologically recalcitrant alkyl and aryl materials largely intact (Henriksen and Breland, 1999; Trumbore et al., 1989); **The E-fraction**: dichloromethane was used for extracting nonpolar extractives (Schnitzer and Schuppli, 1989) (e.g. fatty acids, long-chain alcohols, wax esters, oils, resins etc.); **The N-fraction** is the residue remaining after hydrolysis in sulphuric acid (also known as Klason lignin). See Lines 302-312.

The WAEN protocol is therefore tailored to suit the decomposition models where the recalcitrance to degradation dictates the turnover time of the respective fractions. Besides, as acknowledged by the reviewer this order of decomposability 'W-A-E-N' is coherent with previous plant litter decomposition experiments, which showed that, 'N' is the most stable fraction, then 'E' is relatively more

		stable, and 'W' and 'A' are more labile fractions as their composition amount (%) in total litter mass changed a lot from the initial stage to the end of the decomposition. As suggested by the reviewer, we explicitly referred to our comparison with the plant data and its use in carbon models when introducing the rationale for this method, we added a figure of real plant litter decomposition experiments to support the order of 'W-A-E-N' (see Fig.S5 in supplementary material). Besides, we have modified two paragraphs to better address the concerns of this point, see changes in Lines 301-323.
Reviewer #2 _R2.2.	4: It might be good to define 'chemical recalcitrance' here or to use another word since this term is not universally accepted in the literature. 19: I would also define here what is meant by "quality" - as this could mean chemical recalcitrance or something else	We agree that it is necessary to clarify the words "litter quality", "chemical quality", "chemical recalcitrance" and "chemical decomposability". Reviewer #1 (R1.1.) also had similar concerns. We now refer to chemical proxies of litter decomposability in the revised manuscript text (Line 3, Lines 20-23) and ensured consistent choice of wording on this topic throughout the paper (e.g. by removing 'chemical recalcitrance' throughout, and consistently using 'decomposability' instead) to avoid confusion on the terminology.
Reviewer #2 _R2.3.	21: This is absolutely possible, but it could also be interpreted as lending support to the idea that chemical recalcitrance causes C accrual, which you obviously want to be careful about suggesting. It might be useful to cite some papers discussing how chemical recalcitrance of substrates influences microbial growth efficiency and in turn C storage (e.g. Frey et al. 2013, Nature Climate Change).	Thank you for the suggestion, we have added some information in the discussion and citation to explain our intention to avoid misleading (see Lines 23-24).
Reviewer #2 _R2.4.	25-26: I very much agree.	Thank you for your support!
Reviewer #2 _R2.5.	53-54: Could you specify if this is something you are hypothesizing here or if this is a more general statement. If it is a more general statement, adding some citations would strengthen the argument.	We added a reference to support our hypothesis (Line 59).
Reviewer #2 _R2.6.	Line 63: I'm not super clear what 'integrative' means...could you be more specific? This also applies to line 76.	Thank you for pointing this out. We replaced 'integrative' with 'comprehensive' in Line 68 to avoid confusion. Also applied to Line 84 (previously Line76).

Reviewer #2 _R2.7.	Line 71: It might be better to use a different example from melanin here since not all fungi produce this.	Indeed, melanin is not present in all fungi, but the vast majority of fungi has melanin. We thus rephrased and softened the sentence with 'Most fungi...'. See changes in Line 79 (previous Line 71).
Reviewer #2 _R2.8.	Line 93-34: This hypothesis was somewhat surprising since you did not provide background information in the introduction on this comparison. I would add this background information in order to strengthen this point. This comparison is so interesting, and I really want your future readers to understand the potential significance of it.	Thank you for supporting this comparison part. We further extended the considerations on the comparison of plant litter vs fungal litter in the introduction section (Line 25-29, Lines 75-78).
Reviewer #2 _R2.9.	101-102: I very much respect how hard it is to isolate and culture AMF – but only four species are used here and they are all Glomeraceae. Thus, your results about AMF are limited...it would be good to acknowledge this in the discussion. Other families which produce more extensive biomass in the soil (e.g. Gigasporaceae) or which might be more stress tolerant (e.g. Acaulosporaceae) could have quite different mycelial chemistry.	Thank you for your suggestion. We recognize that it is important to acknowledge in the discussion that our study is limited to the most predominant family of AM fungi (Davison et al., 2020), and does not include members of other families that, although occurring in less relative abundances in nature, have different life-history strategies, and could potentially differ in chemical composition. There are two main reasons why our paper focuses on members of the family Glomeraceae. The first is conceptual and reflects the fact that Glomeraceae is by far the most globally dominant family in AM fungal communities (see, for example, the Fig.6 of Davison et al 2020 New Phytologist, which is the largest global analysis on AM fungal community composition published to date). Therefore most AM fungi-related literature focusses on Glomeraceae; The second reason is practical. The chemical analyses of AM fungi require large amounts of pure fungal biomass. To this end, we are limited to in vitro cultivation to exclude the presence of other microbes. To the best of our knowledge, only a few strains of the Gigasporaceae family have been successfully maintained in vitro across generations, while regarding Acaulosporaceae, only one species has ever been grown in vitro (Acaulospora rhemii) with very limited production and difficult maintenance. We did attempt to include in our study one of the most productive strains of Gigasporaceae (MUCL

		52331) under in vitro conditions. However, even this strain is remarkably slow-growing compared with those of Glomeraceae, and the biomass amounts obtained were extremely low. We thus concluded that it would be unfeasible to produce enough material from this strain within a reasonable time frame. To address the reviewer’s concern, we have now included in the discussion the statement that “Our study focussed on members of the globally most predominant family – Glomeraceae (Davison et al., 2020). It remains to be determined whether less frequent AM fungal families with different life-history strategies potentially differ in their chemical composition” Lines 179-181 (previously in Line 170). Finally, if the working hypothesis raised by the reviewer, i.e. that other AM fungal families have a different chemical composition compared to Glomeraceae, is true, it would re-enforce our main conclusion, i.e. that differences in mycorrhizal decomposability represent an important driver of the soil C cycle, and could be as vital as it is recognized for the differences among aboveground plant litter.
Reviewer #2 _R2.10.	111: You also note an analysis of variance in the methods section - are those results not included?	There might be some misunderstandings following our statements in the original manuscript Lines 111 and 299 (current version Lines 119 and 345). The full name of perMANOVA is the permutational multivariate analysis of variance. We did not include an analysis of the variance of the individual groups.
Reviewer #2 _R2.11.	133: Why do you think you discovered higher A components in AMF biomass? Could it have anything to do with the potentially higher spore volume in the AM vs. EM mycelium? It would also be interesting to acknowledge the species which stand out and why that might be.	We cannot find direct answers to this question, unfortunately. This A component is neither the most recalcitrant nor the most labile and we know very little about what this fraction represents for the ecophysiology of fungal mycelium. So, we need to assume its representation in fungi is similar to that in plant litter and soil organic matter (von Lützow et al., 2007) where it is related to the extraction of carbohydrate and protein materials by disruption of hydrolytic bonding, leaving the more biologically recalcitrant alkyl and aryl materials largely intact (Henriksen and Breland, 1999;

		Trumbore et al., 1989). Why AM fungi have higher amounts of this A fraction or why certain species stand out, we unfortunately do not know yet.
Reviewer #2 _R2.12.	142: I find this framing of mycorrhizal versus plants very interesting, and I like it, so I want others to appreciate it. To make it more generally accepted, I'd like to see you discuss any potential limitations of this comparison. Notably, how does variation in the sample size of plants versus fungi influence these results? If you do not think this has an effect, can you state why you think the results are robust to unequal sample sizes and potential unequal variance? Do you see the same results if you randomly sample from the plants to equal the sample size of fungi? Do you see something similar if you compare the groups using non-parametric techniques (e.g. comparing coefficients of variation across all groups) or multivariate analysis of dispersion? Not all of these need to be tested, I'm just proposing some ideas that could complement the findings beyond the parametric effect size measurements.	Thank you for supporting this comparison and the suggestions for additions to the discussion. We agree with your opinion that there are still limitations of using this data for comparison to plant litter. More information about dispersions and variation across groups of plant litter is now provided in supplementary information (see the update in supplements Fig.S6, Notes.1) and main text. One-way ANOVAs are robust to variation in sample size. Moreover, in this case, the sample size of the fungi is limiting. With more data on fungal decomposability, the statistical power of the analysis would further increase, which should logically lead to even stronger significant effects. In that sense, it is striking that we were able to detect significant differences with our currently low statistical power. We added a discussion to acknowledge the limitations of this comparison: "this comparison of decomposability between mycorrhizal fungi and plant foliage litter is the first attempt at examining such characters of different substrates of decomposition litter. Future research with more fungal species may reveal information beyond the limited data used in this study", see Line 206-209.
Reviewer #2 _R2.14.	170-172: Your results do not entirely support this conclusion. The A compounds are most abundant, and there were higher concentrations in AMF versus EMF tissues. I would therefore be more cautious with this statement as it only applies in relation to the E and N compounds.	We define the 'decomposability' from most labile to most recalcitrant with the order of W-A-E-N, not purely based on chemical structure but also considering the 'turnover rate' in the YASSO model (Liski et al., 2005; Tuomi et al., 2009), which also seems to be representative for natural decomposition processes. We added information to the Supplementary materials to support this order of decomposability W-A-E-N with a figure of real plant litter decomposition dynamics (see Fig.S5 in supplements and changes in the main text in Lines 318-324, also see reply to R2.1.). In our model, water-soluble and acid-soluble are relatively labile to decomposition compared to ethanol soluble and non-soluble components. Thus, according to the

		decomposability order of W-A-E-N, we draw the conclusion that EM fungi have a higher content of recalcitrant components (E and N), likely leading to residues of EM fungi staying longer in the soil as POM; while C loss from AM and EM fungi is enabled through two distinct pathways – AM fungi lose A-components, while EM fungi lose W components. Also if we put aside a potential divergence in opinions about the ‘recalcitrance order’, we show that chemical components in AM and EM fungi are significantly different, and they determine the fate of C during the decomposition of litter.
Reviewer #2 _R2.15.	220: It would help to add a sentence here saying why or why not these fungi growing on different media would or would not affect the results. For example, differences in the media C:N ratio seems like it would differentially influence mycorrhizal biomass chemistry.	We have opted to use standard laboratory cultivation protocols adapted to each species, in contrast to using one common medium to all species, because in this way we assure that each species develops a healthy biological constitution during cultivation to allow comparing the chemical compositions among healthy specimens. The alternative option would be to grow all species in a similar growth medium. However, we have prior knowledge that this is not possible for the different mycorrhizal guilds due to their inherent biological differences. Although EM fungi can grow in vitro on carbon incorporated into the medium, the in vitro cultivation of the obligate biotrophic AM fungi require the presence of compatible AM host roots. Even among EM fungal species, forcing growth in a common standard medium could differently affect each species, as those adapted to the selected medium would likely thrive well, while those less adapted would likely become stressed, fail to grow, or produce sub-optimal constitutions. To address the reviewer’s concern, we have added the following sentence to the materials and methods sections: “We have opted to use laboratory cultivation protocols adapted to each fungal species to assure that each species develops in a most healthy way during mass cultivation. This ensures that the chemical composition is representative for each mycorrhizal type in general, while it may differ from specific local soil

		conditions.” (current Lines 228-231, previous Line 212).
Reviewer #2 _R2.16.	237: I would add a sentence here describing why these different methods were used. Since you say they are “well-developed” in the lab of UCLouvain, it seems unclear why all four approaches needed to be used.	Indeed, the lab of UCLouvain has a rich experience in cultivating AM fungi under in vitro conditions using different techniques, but these cultivations were never aimed to yield high amounts of fungal biomass. We recognized therefore that the chosen adjective “well-developed” does not directly apply to the current purpose, and we removed it from the text. The four approaches chosen are justified by the pioneer aspect of the effort, i.e. generating large amounts of pure biomass of different AM fungi, which are all slow-growing organisms. As mentioned in the main text (now in Lines 244-246), there was no prior knowledge on biomass output among in vitro cultivation approaches’, i.e. we did not know in advance which cultivation system would be the most effective in terms of workload requirements and biomass productivity. Some systems are more productive but require a higher level of work input, while others produce less biomass with less workload. By recruiting a combination of different cultivation characteristics, we were able to optimize productivity within the project’s time frame to successfully assemble the required standard amounts needed for the chemical analyses. To address the reviewer request, we added to the materials and methods section the following statement: “We recruited a combination of different cultivation characteristics to maximize biomass productivity...” (Lines 259-260, previous Line 239)
Reviewer #2 _R2.17.	291-292: I think additional details on these methods and interpretations need to be added to the methods and discussed very clearly in the discussion. As noted above, I particularly argue this because the concept of recalcitrance is centered in the manuscript. I am concerned that some of these methods rely on concepts which have very little biological basis (e.g., as discussed by Lehmann and Kleber, 2015, Nature). For example, these are operationally defined chemical fractions, some of which are well-known to contain	The issues mentioned in this point has also been addressed in R2.1. We agree that the biological impacts of the various chemical fractions are increasingly debated, and individually the extractions have an uncertain ecological meaning. However, together, the components extractable through different chemical methods represents the suite of decomposability related to the extraction capacity of different types

	artefacts from sample preparations (e.g., acid extractions and assuming the non-hydrolysable mass is very refractory is no longer widely accepted). Some of these artefacts have given rise to absurdly long residence times of certain types of compounds in plants (e.g. lignin) which actually decompose within 1 year (e.g., Rasse et al 2006, J. of Soil Science). I encourage you to outline each extraction, and how it can be interpreted based on previous evidence. I would not rely on soil C models for this either since it is these very ways of grouping that has given rise to flawed estimates....many propose we need to move beyond this way of binning C inputs (e.g. Todd Brown et al. 2013, Biogeosciences; Sulman et al 2018, Biogeochemistry). In the discussion, it seems important to acknowledge the limitations of using this classic extraction approach and why more modern spectroscopic methods were not used beyond the one stated sentence.	of enzymes potentially excreted by saprotrophic organisms. Thus we use chemical fractions as a way to operationalize decomposability for which currently no generic alternative methods exist. Besides, spectroscopic methods, although advocated increasingly, have not yet been proven to be generally applicable and interpretable in terms of decomposability. We appreciated your suggestions on how to deal with this, and outlined our solution in the reply to the comment R2.1. In short, we have added to the manuscript a text describing the biochemical meaning of the WAEN fractions (Lines 305-313). In addition, as suggested by the reviewer, we explicitly refer to our comparison with the plant data and its use in carbon models when introducing the rationale for this method, and we added a figure of real plant litter decomposition experiments to support the order of 'W-A-E-N' (see Fig.S5 in supplements). We have modified two paragraphs to better address the concerns related to this issue, see changes in Lines 302-324.
Reviewer #2 _R2.18.	293: From Note S1 and Table S3, adding plant litter seems to have a huge effect that cannot be corrected for in all of the extractions, especially for the non-extractable compounds. Based on Figure 1, the two samples which were mixed with plant litter have the largest N columns...this seems like it is an artifact of the plant mixture. I'm concerned that these two cultures may not be directly comparable to the other species for this reason.	Thank you for your concern about these two cultures which were measured as a mixture. As you mentioned, we have provided the accuracy assessment (previously in the supplementary information, Note.S1, Table.S3, now move to the main text see Lines 327-343, Table.2). According to the data presented in this table, the fractions of WAE were generally properly estimated even with a very small proportion within the plant mixture (with accuracies around 90%). We also agree that the uncertainty in the N fraction is the most pronounced, although our assessment suggests that the N fraction is also convincing when we have more than 60% of mycorrhizal biomass in the mixture. To account for the larger uncertainty in the N fraction, we have taken less weight for these two strains, with only 0.5 instead of 1 as the default weight value for other data (main text Lines 336-340). According to us, this provides the best balance between comprehensiveness and uncertainties.

Reviewer #2 _R2.19.	299: Which distance metric was used for these analyses? If Euclidean distance, were they independent?	The 'distance metric' in our analysis is the 'Bray-Curtis' method using the Vegan package for perMANOVA in R, indicated in Line 347. And the samples themselves were independent of each other.
Reviewer #2 _R2.20.	301: Because the EM and AM sample sizes differ, was this used to test whether variation in perMANOVA results was due to dispersion or differences in centroids (sensu Anderson and Walsh 2001, Ecological Monographs)?	Indeed, given the differences in sample size and the resultant unequal variance, we tested for differences in the dispersion with the 'betadisper()' function in R. The results showed that dispersion was non-significant ($p=0.2937$), indicating that the significant differences in the perMANOVA were due to differences in the mean values. We have added more details about the result of the perMANOVA and the beta dispersion in the main text and supplements, see changes in Lines 119-121, and Table.S3.
Reviewer #2 _R2.21.	310: Were these tests also in R?	Yes, we used the package 'sjstats' in R to perform this Mann-Whitney U test. Now, we added details to the method, see Lines 359 (previous Line 310)
Reviewer #2 _R2.22.	312-315: If you exclude these two species do you detect the same results?	Using only two strains of AM fungi will make the sample size too small (<3) for reliable statistics. However, we agree that the measurements from the mixture should be evaluated carefully. Thus, we have conducted an accuracy assessment for measurements from the mixture across a full range of proportions with standard litter (previously in supplements now included in the main text, Lines 325-343). According to the accuracy assessment, we opted for a weight of 0.5 to be used for these two species measured in a mixture (see main text Lines 336-340), which should account for the uncertainties and eliminate the concerns raised here.
Reviewer #2 _R2.23.	318: Is ANOVA appropriate given the fact that you used non-parametric tests for comparing between EM and AM samples? What makes parametric tests appropriate here but not earlier? I'm also a bit concerned by the larger sample size for plants versus fungi here. This unequal sampling seems like it would potentially affect within group sample variance? Would you consider trying to account for this by adding variance structure statements in your models (e.g. using the varIdent	Above, in the reply to Reviewer #1 _R1.4, we have provided details about our decisions on each analysis in the main text, including the reason why we use non-parametric methods for comparing EM and AM samples throughout our manuscript. In contrast to some parametric methods, non-parametric tests as applied here are relatively robust to sample variance and unequal sampling (by

	function R). Do you also have replicate values for fungi but not for plants? If you have replicate values for both fungi and plants you could also use mixed effects models and use species as a random effect.	the nature of their algorithms). Please note that we used species as replicates in our analyses and thus mixed effect models are less appropriate.
Reviewer #3_R3.0.	Huang and colleagues presented a study about the differences of mycelium decomposability between AM and EM fungi from samples obtained from in vitro cultivation. They found that mycelium decomposability of AM and EM fungi varies among different chemical decomposable substances, having EM fungi a higher concentration of labile and recalcitrant components than AM fungi. I found this manuscript interesting and fulfill a knowledge gap in the soil carbon literature. The experiments are simple and straight forward. However, there are some issues that needs to be addressed/clarified.	Thank you very much for the positive assessment and constructive criticism. Below we address the comments one by one.
Reviewer #3_R3.1.	The first main comment I have is that the manuscript makes a strong emphasis on soils. Yes, I agree that EM and AM fungi are an overlooked component of the terrestrial carbon cycle, however, the mycelium used was grown and harvested from specific media in petri dishes (where is the soil?). This issue has two components that the authors did not discuss: 1) there might be chemical differences between EM and AM fungi grown in media vs grown in soil? 2) those chemical differences might be translated to different decomposability of both EM and AM fungi. In this matter, there might be different solutions, but I suggest to the authors to acknowledge that the results are only “potential” decomposability rates, as well as that the chemical composition of the grown mycelium might be different than grown in actual soil. This might be a substantial change in the text to tone down the global biogeochemical implications.	Thank you for your inquiry. A very similar concern has been posed by Reviewer 2. Therefore, we also refer to our reply to Reviewer #2_R2.15 and changes in Lines 228-231. By opting for using standard cultivation protocols adapted to each species, we were able to compare the chemical compositions among healthy specimens. This approach eliminates impacts of soil-specific differences in conditions. We consider the chemical composition data of WAEN to be representative for each mycorrhizal type in general, while we acknowledge that locally its composition may differ. To avoid local sampling conditions to affect our outcomes, we deliberately chose not to sample them in the field to eliminate this as confounding factor.
Reviewer #3_R3.2.	The second main comment I have is about the comparison between the chemical decomposability traits between EM and AM fungi versus plant litter. I found this comparison out of place in the manuscript and should be removed. There are different reasons for this: 1) the title of the manuscript is about EM and AM fungi, and the decomposition of plant litter is just a final line in the abstract, with no substantial results; 2) in the	Thank you for your suggestion. We agree that not enough background for plant litter was provided in the original manuscript. But we think it is still worthwhile to have this comparison, see comments Reviewer #2_R2.8 and R2.12. Thus, we added more background for the reason why we made the comparison to plant litter, see lines 29-30, 74-76.

	introduction section it is only mentioned in the hypotheses (!), so there is not background on why this comparison is made or why this is important, and it is discussed in the results/discuss section; 3) data for the litter decomposition come from two different known databases, with different sample sizes for the plant functional types, making a no-fair comparison.	Concerning the variation in the sample size of plants versus fungi, Reviewer #2 had a very similar concern. Please find our reply in the R2.12. We have added a discussion to acknowledge the limitations of this comparison “This comparison of decomposability between mycorrhizal fungi and plant foliage litter is the first attempt to compare these two important sources types of carbon input into soil. Future research with more fungal species may reveal information beyond the limited data used in this study”, see Line 206-209.
Reviewer #3_R3.3.	My third main comment is related to how the authors reported the sample sizes and how the statistics were done. 1) It is not clear which is the real sample size of EM and AM fungi for stats, seems that 600 for AM but where is the value of EM?; the authors used the fungal guilds as replicates (n = 4 for AM and n = 11 for EM), are the authors using an average value from averages values? If so, why not using the raw data pooled together as AM or EM? Using an average of averages smooths a lot of variability, and if the sample size is different the Mann-Whitney test might be still used. 2) differences presented in figure 3 (boxplots) come from the log (x+1) transformed of, again, n = 4 for AM and n = 11 for EM, arguing that the log transformation is due to lack of normality (are we expecting normality with an n=4?); I suggest to the authors to report the results from a normality test; however, I strongly suggest using the raw data and not using the fungal guilds as replicates. This should be also reflected in submitting all the raw data to a repository and not only reporting the “limited” data in the supplementary material. Another potential solution could be reporting in supp mat boxplots of the different fungal guilds so the reader can see that there is a difference between EM and AM fungi by guilds.	There seem to be some misunderstanding here. (1) What we present in the results are ‘raw data’ (not averaged). We have 11 samples of EM and 4 samples of AM fungi, fungal guilds are replicates (Lines 121-122). Samples of each strain of AM and EM fungi were collected from multiple systems/plates. For the number of ‘600’ mentioned in Line 261, it is the approximate number of systems/plates established for AM fungi. The samples for these plates had to be merged to obtain enough sample material for analysis, ending up with one sample per species. Thus, there is no raw data from the individual plates, yet these data would be a pseudoreplication in the framework of our analysis, which used fungal species as true replicates. This is all the data we have (due to the fact that cultivation for getting enough material is very hard). (2) We tried a log-transformation to mitigate heterogeneity of variance before conducting analysis, but that did not solve the heterogeneity issues. Thus we have redone these two non-parametric analyses using the original data. However, it does not influence our main results and conclusions. Please see the update in the methodology (Lines 347-349), Fig.2 and Fig.3 (also see the feedback to comments Reviewer 1#_R1.4.).

References

- Anderson, M. J.: A new method for non-parametric multivariate analysis of variance, *Austral Ecol.*, 26(1), 32–46, doi:10.1111/j.1442-9993.2001.01070.pp.x, 2001.
- Anderson, M. J.: *Permutational Multivariate Analysis of Variance (PERMANOVA)*, Wiley StatsRef Stat. Ref. Online, 1–15, 2017.
- Davidson, E. A., Galloway, L. F. and Strand, M. K.: Assessing available carbon: Comparison of techniques across selected forest soils, *Commun. Soil Sci. Plant Anal.*, 18(1), 45–64, doi:10.1080/00103628709367802, 1987.
- Davison, J., García de León, D., Zobel, M., Moora, M., Bueno, C. G., Barceló, M., Gerz, M., León, D., Meng, Y., Pillar, V. D., Sepp, S. K., Soudzilovskaia, N. A., Tedersoo, L., Vaessen, S., Vahter, T., Winck, B. and Öpik, M.: Plant functional groups associate with distinct arbuscular mycorrhizal fungal communities, *New Phytol.*, 226(4), 1117–1128, doi:10.1111/nph.16423, 2020.
- Henriksen, T. and Breland, T.: Evaluation of criteria for describing crop residue degradability in a model of carbon and nitrogen turnover in soil, *Soil Biol. Biochem.*, 31(8), 1135–1149, doi:10.1016/S0038-0717(99)00031-0, 1999.
- Liski, J., Palosuo, T., Peltoniemi, M. and Sievänen, R.: Carbon and decomposition model Yasso for forest soils, *Ecol. Modell.*, 189(1–2), 168–182, doi:10.1016/j.ecolmodel.2005.03.005, 2005.
- von Lütow, M., Kögel-Knabner, I., Ekschmitt, K., Flessa, H., Guggenberger, G., Matzner, E. and Marschner, B.: SOM fractionation methods: Relevance to functional pools and to stabilization mechanisms, *Soil Biol. Biochem.*, 39(9), 2183–2207, doi:10.1016/j.soilbio.2007.03.007, 2007.
- Schnitzer, M. and Schuppli, P.: Method for the Sequential Extraction of Organic Matter from Soils and Soil Fractions, *Soil Sci. Soc. Am. J.*, 53(5), 1418–1424, doi:10.2136/sssaj1989.03615995005300050019x, 1989.
- Trumbore, S. E., Vogel, J. S. and Southon, J. R.: AMS ¹⁴C Measurements of Fractionated Soil Organic Matter: An Approach to Deciphering the Soil Carbon Cycle, *Radiocarbon*, 31(03), 644–654, doi:10.1017/S0033822200012248, 1989.
- Tuomi, M., Thum, T., Järvinen, H., Fronzek, S., Berg, B., Harmon, M., Trofymow, J. A., Sevanto, S. and Liski, J.: Leaf litter decomposition—Estimates of global variability based on Yasso07 model, *Ecol. Modell.*, 220(23), 3362–3371, doi:10.1016/j.ecolmodel.2009.05.016, 2009.
- Xu, G., Long, Z., Ren, P., Ren, C., Cao, Y., Huang, Y. and Hu, S.: Differential responses of soil hydrolytic and oxidative enzyme activities to the natural forest conversion, *Sci. Total Environ.*, 716, 136414, doi:10.1016/j.scitotenv.2019.136414, 2020.

REVIEWERS' COMMENTS:

Reviewer #1 (Remarks to the Author):

The authors have addressed all of my concerns.

Reviewer #2 (Remarks to the Author):

My previous comments have been fully addressed in the revision. I am still not super on-board with using these chemical extraction operational groupings as proxies for decomposition in the absence of other decomposition metrics. It raises the possibility of repeating very well-known issues of applying these metrics to plant litter decomposition but for fungal necromass. Nonetheless, the authors have presented the data as clearly as they can, and I appreciate that they have added additional explanation of the potential meaning of these groups.

Reviewer #3 (Remarks to the Author):

Huang and colleagues presented a revised manuscript about the differences of mycelium decomposability between AM and EM fungi from samples obtained from in vitro cultivation. T

I found the revised version of the manuscript well written and overall, my comments/suggestions, as well as the other reviewers comments/suggestions have improved the manuscript clarity. I strongly believe that this manuscript fullfill an important knowledge gap in soil C literatute. I do not have more comments/feedback to the authors, and I really appreciate the time they took in solving the issues arised.